# Nerve Structure-Function: Unusual Structural Details and Unmasking of Sulfhydryl Groups by Electrical Stimulation or Asphyxia in Axon Membranes and Gap Junctions

**DOI:** 10.3390/ijms241713565

**Published:** 2023-09-01

**Authors:** Camillo Peracchia

**Affiliations:** Department of Pharmacology and Physiology, School of Medicine and Dentistry, University Rochester, Rochester, NY 14642-8711, USA; camillo.peracchia@gmail.com or camillo_peracchia@urmc.rochester.edu

**Keywords:** axons, nerves, Golgi apparatus, plasma membrane, endoplasmic reticulum, mitochondria, crayfish, gap junctions, sulfhydryl groups, electrical stimulation, asphyxia

## Abstract

This review describes and discusses unusual axonal structural details and evidence for unmasking sulfhydryl groups (-SH) in axoplasmic membranes resulting from electrical stimulation or asphyxia. Crayfish axons contain fenestrated septa (FS) that, in phase contrast, micrographs appear as repeated striations. In the electron microscope, each septum is made of two cross-sectioned membranes containing ~55 nm pores, each occupied by a microtubule. Thin filaments, which we believe are made of kinesin, bridge the microtubule to the edge of the pore. FS are believed to play a role in axoplasmic flow. The axons also display areas in which axon and sheath glial cell plasma membranes are sharply curved and project into the axoplasm. In freeze-fractures, the protoplasmic leaflet (P-face) of the projections appears as elongated indentations containing parallel chains of particles. The sheath glial cell plasma membrane also contains particles, but they are irregularly aggregated. The axons also display areas where axonal and glial plasma membranes fuse, creating intercellular pores. In axons fixed during electrical stimulation, the plasma membrane, the outer membrane of mitochondria, membranes of other cytoplasmic organelles, and gap junctions increase in electron opacity and thickness, resulting from unmasking of sulfhydryl groups (-SH). Similar changes occur in asphyxiated nerve cords.

## 1. Introduction

This review brings back to light the unusual structural details of crayfish axons [1,2,3,4], some of which are also present in vertebrate axons, and describes changes in axoplasmic membranes resulting from electrical stimulation or asphyxia; these changes are characterized by a drastic increase in electron opacity of certain axonal membranes resulting from the unmasking of sulfhydryl groups (-SH) in membrane proteins [2]. We feel that these data are important and need to be further studied, confirmed, and interpreted in detail.

## 2. Structural Details of Crayfish Axons

### 2.1. Fenestrated Septa

Axons sectioned longitudinally or slightly obliquely and viewed by phase-contrast microscopy display regularly spaced striations (Figure 1A). In electron micrographs, the striations correspond to parallel fenestrated septa (FS) that cross the axoplasm perpendicularly to the long axis of the axon (Figure 1B–D) [1]. FS are cisternae whose membranes are separated by a narrow gap (Figure 1B–D). The membranes of FS frequently join, forming discontinuities (fenestrae) that are crossed by microtubules (Figure 1C, arrows). Neighboring FS are joined by membranous tubules (Figure 1D).

In cross-sectioned axons, the FS’s discontinuities appear as 50–60 nm pores, each occupied by a microtubule (Figure 2A–C). In some pores, thin filaments ~10 nm long link the microtubule to the edge of the pore (Figure 2C). We believe that the filaments are kinesin molecules [5], which indeed are ~10 nm long. The heavy chain of the crayfish kinesin, a 120 kDa protein, was isolated in 1996 by Okada and coworkers [6] and named CF-kinesin. We have reported the presence of similar structures in mouse myelinated axons [1,5], which indicates that they are present in all animals. We have proposed that FS are elements of the anterograde transport of cisternae of the Golgi apparatus mediated by motor proteins, which we believe to be kinesins [5], although this needs to be proven.

### 2.2. Structural Interaction between Axons and Sheath Glial Cells

Cross-sectioned and cross-fractured crayfish axons display regions in which the plasma membranes of axon and sheath glial cells are regularly curved and project into the axoplasm (Figure 3A–C) [3]. At these regions (projections), the two plasma membranes run precisely parallel, separated by a 13–14 nm gap (Figure 3B). In longitudinal freeze-fracture replicas, the axons display the inner fractured faces of either the internal (protoplasmic, P-face, APF) or the external (exoplasmic, E-face, AEF) leaflet of both axonal and adjacent sheath glial cell plasma membranes (Figure 3D). In both the axonal protoplasmic face (APF) and exoplasmic face (AEF) (Figure 3D), the projections are seen as elongated structures oriented with their long axis parallel to the long axis of the axon (Figure 3D). On the P-face of the axon’s plasma membrane (APF), the elongated indentations are 0.5–1.2 μm long and 0.12–0.15 μm wide (Figure 3D, arrowheads). The E-face (AEF) of the axon’s plasma membrane displays elongated protrusions, which are the complementary images of these structures (Figure 3D, arrows).

The axonal indentations contain parallel chains of ~8 nm particles (Figure 4A) that repeat every 12–12.5 nm and are obliquely oriented such that in the axonal P-face (APF), the chains’ axis (b in Figure 4C) is skewed with respect to that of the indentation (a in Figure 4C) by an angle of 55–60° (Figure 4A,C,D). The particles repeat along the chain every 8–8.5 nm (Figure 4A). The freeze-fracture E-face (AEF) shows complementary chains of pits (Figure 4B). The particles of one chain are in register with those of the adjacent chain and are aligned along “axis c” (Figure 4C), which forms an angle of 75–85° with “axis b” of the chains (Figure 4C). This complex structure forms an array with a rhomboidal unit cell of 80–85 × 12–12.5 nm (Figure 4A,C).

The projections of the sheath glial cell’s plasma membrane also contain particle arrays (Figure 4E,F). Still, these arrays differ from those in the axonal projections in size, pattern of aggregation, and fracture properties. Indeed, while in freeze-fracture replicas, all the axonal particles remain with the protoplasmic leaflet (Figure 4A,B), in the sheath glial cell, some particles are with the protoplasmic leaflet (Figure 4E) and others with the exoplasmic leaflet (Figure 4F). It is hard to propose a functional meaning for these structures; they could be areas of cell–cell adhesion, regions of metabolic couplings, or other unknown types of cell–cell interaction [3].

### 2.3. Membranous Pores Joining Axons to Sheath Glial Cells

In certain areas of contact between axons and sheath glial cells, the two plasma membranes fuse, forming membranous pores (Figure 5 and Figure 6) [4,7]. Several pores are usually grouped in the same area (Figure 5). Often, in these areas, one sees aggregates of endoplasmic reticulum (ER) or Golgi cisternae, mitochondria, and other membrane structures (Figure 6). Most pores are 15–20 nm in size (Figure 5), but larger pores are also occasionally seen (Figure 6).

Pores of this size would be expected to allow wide communication between axoplasm and sheath glial cell cytoplasm. Indeed, the existence of pores between axons and sheath glial cells seems reasonable because only passageways of this size would allow the exchange of molecules such as RNA and proteins as heavy as 200,000 Daltons [8,9,10,11]. Viancour et al. [12] reported that when crayfish median giant axons are intracellularly injected with Lucifer Yellow CH, this fluorescent dye rapidly diffuses from the axoplasm to the cytoplasm of the adjacent glial cells; the images they published were startling as they showed that the nuclei of the glial cells bordering the giant axon were all stained by the fluorescent dye [12]. Probably, the pores form transiently but are unlikely to be a preparation artifact because the radius of membrane fusion at the pores (Figure 5 and Figure 6) is consistent with the minimum curvature radius of biological membranes. The axons’ membranes and other cellular components were very well fixed by vascular perfusion with glutaraldehyde followed by osmium tetroxide [4].

## 3. Structural Changes in Electrically Stimulated Axons

### 3.1. Historical Background

Most data on changes in nerve fibers during electrical activity came from electrophysiological studies of axonal membrane channels published in the early-to-mid 20th century. The earliest, definitive evidence of the existence of membrane channels resulted from the work of Alan Lloyd Hodgkin (1914–1998) and coworkers [13,14,15,16,17]. This major finding was enabled by the discovery of the giant squid axon [18] and the inventions of both microelectrodes by Ida Henrietta Hyde and the voltage-clamp technique [19]; rev. in [20]. However, in the last two decades, some questions have been raised about the validity of the Hodgkin–Huxley model. Nerve impulse generation and propagation are frequently believed to be entirely electrical events, but the Hodgkin–Huxley model cannot account for evidence of non-electrical phenomena that accompany nerve impulse propagation [21,22]. Indeed, over the years, several reports on morphological changes, including our own [2], have been published, as mechanical changes have been reported to be associated with excitation in many cells and tissues [23]. For instance, in certain plant cells, excitation is accompanied by outer displacements of the plasma membrane that coincide with the depolarization phase of the action potential, which can be explained by reversible changes in the mechanical properties of the cell surface, such as transmembrane pressure, surface tension, and bending rigidity [23]. Taken together, these findings contribute to the ongoing debate about the physical nature of cellular excitability [21,22,23].

In our study, published over a half-century ago [2], we reported ultrastructural changes in axonal membranes of crayfish nerve fibers resulting from electrical stimulation and asphyxia. The changes are characterized by the drastic increase in osmiophilia and thickness of the axonal plasma membrane and the membranes of cytoplasmic organelles such mitochondria (outer membrane only), ER and Golgi apparatus, and gap junctions; in contrast, the membranes of the sheath glial cells and the mitochondrial inner membrane do not change (see in the following). To our knowledge, our data have not yet been confirmed, although several studies have reported various axonal changes in electrically excited vertebrate and invertebrate axons.

Early in the 20th century, changes in neurofibril staining were reported in axons subjected to polarizing current [24]; these changes were later confirmed in live, unanesthetized nerves [25]. Almost two decades later, changes in light scattering were recorded in axons stimulated in vitro and were believed to result from changes in axonal volume [26,27,28,29,30]. Changes occurring at the speed of the action potential were also recorded [31,32,33] by light scattering and birefringence measurements in stimulated axons; these changes occurred synchronously with the action potential [31]. An increase in fluorescence was also reported in axons treated with 8-anilinonaphthalene-1-sulfonic acid (ANS) [33]. ANS interacts with macromolecules’ hydrophobic groups [34] and changes in fluorescence properties due to conformational modifications in macromolecules [35]. Therefore, these changes were thought to result from excitation-induced conformational modifications in membrane macromolecules. Axons stained with various fluorochromes responded to electric stimulation with transient fluorescence changes, suggesting a drop in membrane viscosity [36].

Several studies have reported initial heat developing with a single impulse in mammalian unmyelinated nerves [37,38], as well as nerve swelling during the action potential [39], but it is not clear whether the heat production in nerves during action potential propagation is reversible [40,41]. There is currently no widely accepted consensus about the mechanism of heat generation, but a dimensionless mathematical model, coupling action potential to mechanical waves with temperature effects, has been proposed [42]. In crabs, nerve impulses produced an outward displacement of the nerve surface and an increase in swelling pressure—this was smaller in squid giant axons [38]. Other studies also reported that in crustacean nerves, the electrical impulse is associated with a small and rapid volume expansion of fibers, which corresponds to increased swelling pressure; tetrodotoxin and procaine suppress the rapid mechanical changes [43]. In squid axons, mechanical changes with the action potential were further studied by piezoelectric and optical methods; the peak of axonal swelling coincided with the peak of the action potential [44]. Repeatedly fired action potentials, caused by reducing external calcium concentration ([Ca^2+^]_o_), resulted in gradual axonal swelling [44]. Transient axonal shortening, followed by elongation, was also reported to occur during action potential [44]. With optical and mechano-electric detectors, the squid giant axon swelled when an action potential was generated, the maximum swelling occurring at the peak of the action potential [45]. These mechanical changes were also seen in axons where much of the axoplasm had been removed [46]. In the garfish olfactory nerve, swelling of the nerve fibers was also reported to occur simultaneously with a shortening of the fibers as an impulse flowed along the fiber [47]—heat production also occurred simultaneously with the action potential. It was proposed that mechanical and thermal changes are related to the release and re-binding of calcium ions during the action potential [47]. Rapid volume expansion of fibers with an electrical impulse has also been reported in unmyelinated axons; this volume expansion resulted from lateral expansion of the excited portion of the axon [48]. In the garfish olfactory nerve, the time-course of swelling and birefringence changes have been studied by applying various chemicals in the place where the electric current pulses were applied—this study demonstrated that a pulse causes a rapid increase, followed by a slow, gradual increase, in axonal water-content [49].

### 3.2. Increased Electron Density (Osmiophilia) in Membranes of Electrically Stimulated Axons

In control preparations, the membranes of crayfish axons are all the same thickness and electron density (Figure 7). In contrast, in crayfish axons fixed during electrical stimulation, the membranes of stimulated axons display a drastic increase in electron density (osmiophilia) and thickness [2]. These changes affect the plasma membrane, the membranes of ER/Golgi cisternae, the outer membrane of mitochondria, and those of gap junctions (Figure 8, Figure 9, Figure 10 and Figure 11), while the inner membrane of mitochondria and the membranes of the sheath glial cells (Figure 8, Figure 9, Figure 10 and Figure 11) are not affected. The increase in electron density and thickness is seen in membranes of the median and lateral giant axons as well as in those of many medium- and small-sized axons, but not all axons are reactive. Indeed, several axons appear as in control samples (Figure 8A and Figure 9B), probably because they were not electrically stimulated.

At higher magnification, the increase in thickness and electron density of the affected membranes is more pronounced on the cytosolic side (Figure 9A and Figure 10). The overall thickness of the membranes increases from 8–8.5 nm to 12–15 nm (Figure 9A and Figure 10). The electron density of the affected membranes is greater than that of any other cellular material, even in unstained sections (Figure 10B).

In some mitochondria, the outer membrane appears granular due to the presence of electron-dense particles (Figure 11A). In cross-sections, the particles are ~15 nm in size and repeat every ~20 nm (Figure 11B). Significantly, these particles are the same size and spacing of gap junction channels (Figure 11C, c), suggesting that they might be gap junction hemichannels.

During our 1960s work on unstimulated crayfish axons fixed by conventional glutaraldehyde/osmium-tetroxide, the gap junction membranes did not display electron-dense particles (Figure 11C, a and b) [50]. It was a lucky coincidence, therefore, that the increased electron density of gap junction membranes caused by electrical stimulation allowed us to reveal for the first time images of gap junction channels (Figure 11C and inset c) [2]. Eventually, we succeeded in revealing gap junction channel images even in unstimulated axons because of glutaraldehyde-H_2_O_2_ fixation [50,51].

The increased osmiophilia is seen after prolonged stimulation as well as after stimulation as short as 30 s. The electron-dense material is not seen in specimens fixed with glutaraldehyde alone and is unstained [2], suggesting that it does not have intrinsic electron density in the absence of osmium tetroxide fixation.

### 3.3. Increased Electron Density (Osmiophilia) in Membranes of Asphyxiated Axons

In ganglia asphyxiated by immersion in saline solution through which CO_2_ or N_2_ is bubbled, the axonal plasma membrane, the membrane of the ER/Golgi cisternae, the outer mitochondrial membranes, and gap junctions display the same increase in electron opacity and thickness as the membranes of electrically stimulated axons [2]. However, in asphyxiated ganglia, all the axons display membranes with increased electron opacity. In asphyxiated axons, as in stimulated axons, the intensity of the reaction is more pronounced in the larger axons, and both the inner membrane of mitochondria and the membranes of sheath glial cells are not affected [2].

### 3.4. Increased Electron Opacity (Osmiophilia) in Membranes of Axons Treated with Sulfhydryl (-SH) Reagents

As a hypothesis, we considered the possibility that the increased membrane osmiophilia of electrically stimulated or asphyxiated axons might have resulted from unmasking sulfhydryl groups (-SH) in membrane proteins [2]. To test it, we treated unstimulated axons with -SH reducing agents, such as 1 h treatment with 0.4 M sodium thio glycolate (TG) or 5 mM dithioerythritol (DTE) in 0.1 M phosphate buffer (pH 7) at room temperature. Significantly, with these treatments, the axonal membranes increase in electron opacity and thickness as in the electrically stimulated axons (Figure 12); as in asphyxiated nerve cords, all the axons are reactive. In specimens treated with -SH reducing agents followed by treatment with -SH blockers, such as 30 min to 1 h exposures to 0.1 M solutions of maleimide or N-ethyl-maleimide (NEM) in 0.1 M phosphate buffer (pH 7.6) at room temperature, the rise in electron opacity and thickness is not observed, as all of the axonal membranes are as in control axons (Figure 13) [2].

## 4. Why Does Membrane’s Osmiophilia Increase in Electrically Stimulated Axons?

Membrane osmiophilia results from the reaction of both lipids and proteins with osmium tetroxide. In membrane lipids, osmium tetroxide reacts with the double bonds of the hydrocarbon chains, forming cyclic osmic esters; reviewed in [52]. The reactions between proteins and osmium tetroxide are more complex. Several amino acids, including diamino acids, sulfur-containing amino acids, and amino acids containing cyclic groups, have been reported to react with osmium [53], but when incorporated into proteins, only a few react with osmium. Free -SH groups of different molecules, such as cysteine and glutathione, readily react with osmium and precipitate, while disulfide bonds (S-S) are much less reactive [53]. In this reaction, the sulfhydryl groups reduce osmium tetroxide to insoluble compounds [53].

While most cellular components are weakly stained by osmium, in some cases, a very electron-opaque osmium precipitate known as “osmium black” is generated. Osmium black results from the reactions between osmium tetroxide and certain reagents containing -SH groups, such as thiocarbamoyl, diazo-thioether, and azo-groups (azo indoxyls) [54]. Studies on the chemical composition of osmium black have reported that it is made of osmium polymers representing coordination compounds of osmium with organic sulfur ligands [55]. Reagents containing -SH groups are the most reactive in the synthesis of osmium polymers, which are insoluble in solvents used for dehydration as well as in embedding chemicals such as acrylic and epoxy compounds. This property is what renders osmium black extremely electron-dense in thin-sectioned specimens.

The fact that osmium black can be synthesized by molecules containing -SH groups strongly suggested to us that the unmasking of -SH groups in membrane proteins could explain the observed increase in membrane osmiophilia. Indeed, our experiments with sulfhydryl reagents strongly support this hypothesis because, with exposure to thioglycolate or DTE, the axonal membranes increase in osmiophilia with the same characteristics and distribution as those in electrically stimulated axons. Moreover, the same membranes appear as in control axons if the newly unmasked -SH groups are blocked by maleimide or NEM before osmium fixation. Osmium reduction has also been reported to occur during zinc iodide-osmium tetroxide (ZIO) fixation, resulting in zinc osmate associated with Ca^2+^ high-affinity sites [56]. Ca^2+^ is necessary for nerve impulse activity and plays an important role in the mechanism determining the specific conductance changes in the nerve membrane [57]. Furthermore, a redox-dependent thiol switch might affect a nearby Ca^2+^-binding site [58], and it has been proposed that structural changes induced by disulfide bond formation may interfere with Ca^2+^-binding, leading to functional consequences such as enzyme inactivation [59].

Several cysteines are known to be present in ion channels of excitable membranes [60]. Four cysteines have been found to be expressed within the pore sequence of Na^+^ channels and multiple others in different domains on these channels, and sulfhydryl modification has been shown to affect channel gating [60]. In invertebrate gap junctions, the channel protein innexin-1 contains four cysteines in the cytoplasmic loop and one in the C-terminus [61]; it is possible that some of these form disulfide bonds that are split into -SH groups by electrical stimulation, asphyxia or treatment with reducing agents.

Electrical stimulation has been reported to affect -SH groups in squid giant axons, whose -SH groups are unmasked by electrical stimulation, and disulfide groups have been found to play a role in conformational changes induced by electrical stimulation [62]. The relevance of -SH groups in the function of excitable membranes is supported by several publications [63,64,65,66,67,68,69,70]. Exposure to -SH reagents has been reported to cause membrane depolarization [71], and the normally unexcitable ventro-abdominal flexor muscles of the crustacean Atya lanipes generated trains of calcium action potentials after treatments with sulfhydryl reagents [69]. Significantly, Baumgold and coworkers [72] have suggested that -SH groups of proteins located on the axoplasmic side of the axonal plasma membrane may play a role in maintaining excitability because modification of these -SH groups blocks axonal conduction.

All the above is consistent with the idea that electrical stimulation causes the unmasking of -SH groups in the excitable plasma membranes. Still, it is quite puzzling why cytoplasmic membranes such as the outer mitochondrial and the membranes of the ER/Golgi cisternae, which are not directly linked to the plasma membrane, also react by increasing in osmiophilia. Puzzling, as well, is why treatments with reduction agents affect axonal membranes while they do not affect membranes of sheath glial cells. An interesting model was put forward in 1991 by Marinov [73]. In this model of channel function and regulation, labile electrons in channel-forming proteins are believed to act as electric field sensors activated by various agents. This model suggests that thiols in oxidized and reduced form may act as redox centers.

In any event, our data prove that within, or very near, the two dense leaflets of the axonal plasma membrane, the outer mitochondrial membrane, the membranes of the ER/Golgi cisternae, and gap junctions of crayfish axons, there are proteins rich in sulfur whose -SH groups are unmasked by electrical stimulation. The fact that the leaflet facing the axoplasmic in each of the reactive membranes appears thicker than the other could mean that it contains a greater density of -SH groups available for reaction with osmium. In this respect, it is noteworthy to remember that also, in membranes of the sarcoplasmic reticulum -SH groups were found more numerous in the homologous membrane leaflet [74]. It has been suggested that these -SH groups may belong to ATPase molecules since treatment with ATP proved to prevent the -SH blocking effect of NEM. In 1983, Lin and Ayala [75] suggested that protein conformational changes occurring when free -SH groups are bound by -SH reagents very likely involve either the release of membrane-bound Ca^2+^ [76] or the inhibition of membrane binding of Ca2+ [77], resulting in altered membrane function; see [78].

In crayfish axons, the -SH-rich proteins could also represent ATPases. Indeed, a Mg-dependent ATPase sensitive to -SH reagents has been found in crustacean nerve microsomes (ER fragments) [79]. Unmasking of -SH groups is believed to cause conformational changes in proteins, which may be relevant to ion transport and membrane electrical properties during excitation. Zuazaga and coworkers [69,70] have suggested that crustacean muscle excitability is due to the conversion of certain CH2-SH side chains to thioethers with carbonyl groups, which may interact with neighboring amino groups and form bonds between different protein domains or protein units. This might affect excitability by inducing conformational changes or by preventing the occurrence of them. This scenario should definitely be kept in mind, but in our case, it seems unlikely because it may not explain the increased osmiophilia that results from the unmasking of -SH groups by treatment with TG or DTE.

Our data show an increase in osmiophilia in the same membranes also because of asphyxia. This may suggest that asphyxia is associated with the reduction of disulfide bonds due to changes in the oxidation-reduction potential. Whether changes in the oxidation-reduction potential sufficient to produce the phenomenon could also occur in electrically stimulated axons cannot be stated. Still, this possibility must be kept in mind.

## 5. Conclusions

The major goal of this review is to bring back to light unusual structural details of crayfish axons and changes in osmiophilia of axonal membranes caused by electrical stimulation, asphyxia, or treatment with sulfhydryl reagents for stimulating more research on the functional meaning of these early findings. Some of the structural details, such as fenestrated septa (FS), are also present in mammalian axons and are probably involved in the mechanism of axoplasmic transport. Other details, such as axo-glial pores, may be transient, and if they are real structures, as we believe, they are probably involved in the transfer of large molecules such as RNA and proteins as heavy as 200,000 Daltons [8,9,10,11], but these and structural interactions between axons and sheath glial cells have not been described in other studies and may not be present in vertebrate axons. Perhaps future investigation should at first determine if vertebrate myelinated and/or unmyelinated axons also develop pores under certain conditions, possibly by injecting radioactive tracers or fluorescent dyes in a neuronal cell body and see if the tracer appears in the Schwann cells, as done in crayfish [12]. To our knowledge, the unmasking of -SH groups in axonal membranes resulting from stimulation or asphyxia has not been reported in other studies. In conclusion, we are fully aware that this small review provides more questions than answers. But, once again, its major goal is to stimulate more advanced research to achieve a better understanding of the functional meaning of these structural findings.

## Figures and Tables

**Figure 1 ijms-24-13565-f001:**
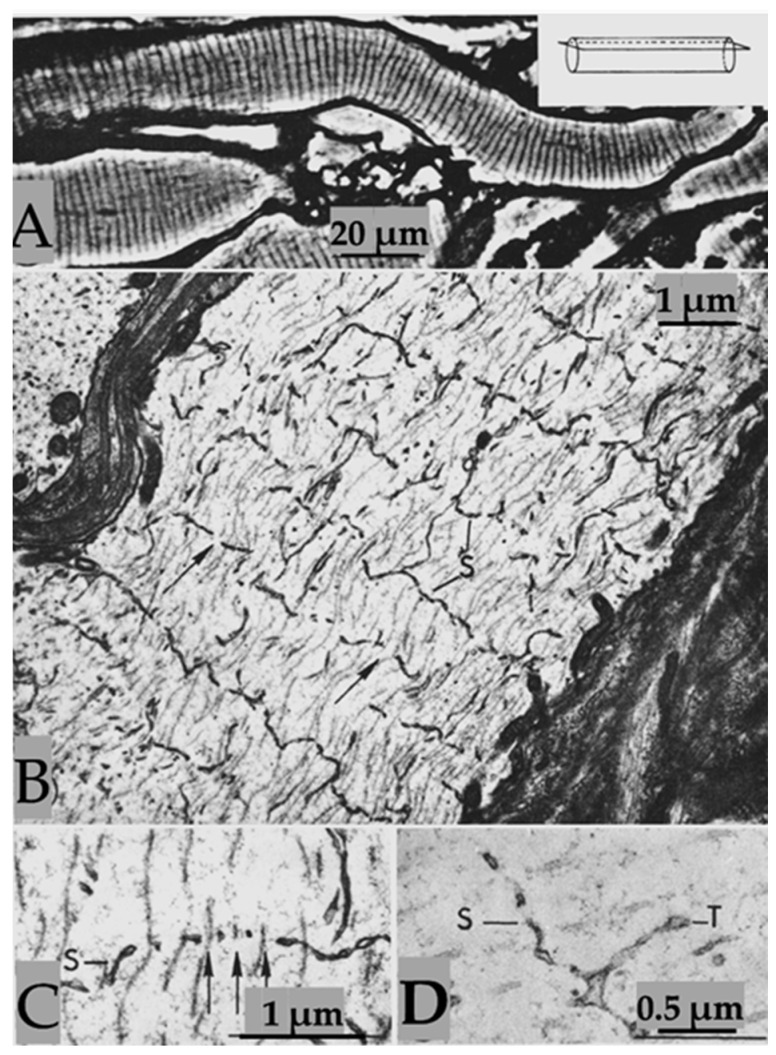
(**A**). Phase-contrast micrograph of medium-sized axons, sectioned in peripheral areas (see inset), displaying parallel septa. Electron micrographs (**B**–**D**) show that each septum is made of two cross-sectioned membranes arranged perpendicularly to the long axis of the axon. The two membranes join, forming 0.1–0.2 μm pores, each crossed by microtubules ((**C**), arrows). Neighboring septa linked by membranous tubules (**D**). S—Septum; T—Tubule. Adapted from Ref. [1].

**Figure 2 ijms-24-13565-f002:**
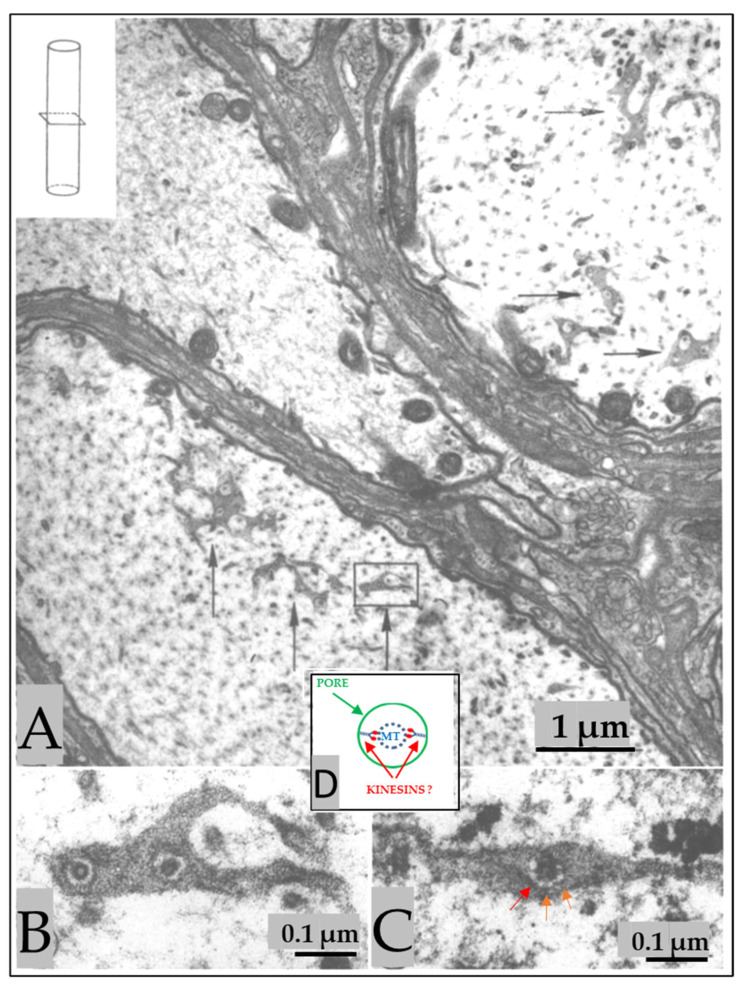
(**A**). Electron micrograph of cross-sectioned axons (see inset in (**A**)) display face view images of the septa shown in Figure 1. The septa, named fenestrated septa (FS), contain pores ((**A**), arrows), each crossed by a microtubule. (**B**). Detail of (**A**) showing two ~55 nm pores, each crossed by a microtubule; the gap between pore-edge and microtubule is 8–10 nm. Frequently, pores display thin filaments ((**C**), red/orange arrows) that link the microtubule to the edge of the pore. We believe that the filaments are kinesin molecules (**D**). Adapted from Ref. [1].

**Figure 3 ijms-24-13565-f003:**
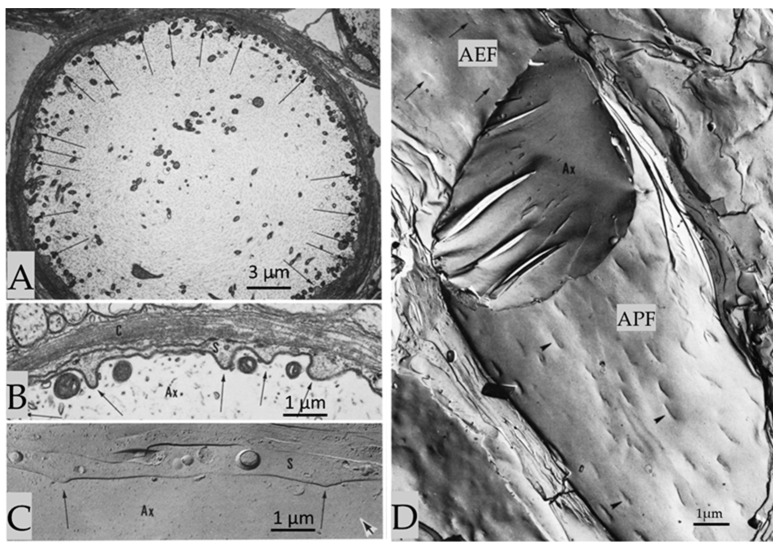
Crayfish axons display specialized regions of interaction with sheath glial cells, where the two plasma membranes project into the axoplasm ((**A**–**C**), arrows), separated by a gap of 13–14 nm (**B**). In freeze-fracture replicas, the P-face of the axonal projections ((**D**), APF) appears as an elongated indentation 0.5–1.2 µm long and 0.12–0.15 µm wide ((**D**), arrowheads). The axonal E-face ((**D**), AEF) shows complementary images of the indentations ((**D**), AEF, arrows). Adapted from Ref. [3].

**Figure 4 ijms-24-13565-f004:**
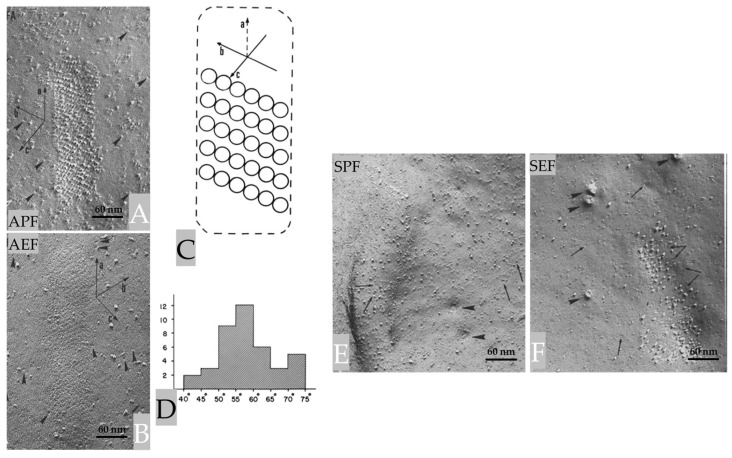
The elongated indentation, shown in Figure 3, contains parallel chains of ~80 nm particles that repeat every 12–12.5 nm ((**A**), APF). The particles repeat along the chain every 8–8.5 nm (**A**). Random particles ~8 nm in size (arrowheads) are seen in both APF (**A**) and AEF (**B**). Complementary arrays of pits are seen in the E-face ((**B**), AEF). This structure has a rhomboidal unit cell of 8–8.5 × 12–12.5 nm (**C**). (**D**), shows a histogram of the frequency of the angle between axes a and b (**A**,**C**). In sheath glial cells (**E**,**F**), the indentations display randomly arranged particles 10–12 nm in size and rarely pits (arrows). In (**E**,**F**), the arrowheads point to dimples and complementary protrusions, ~40 nm in size, representing the openings of the membranous lattice present in the cytoplasm sheath glial cell cells cytoplasm (see in the following). APF and AEF, Axonal P and E fracture faces, respectively. SPF and SEF, Sheath glial cells’ P and E fracture faces, respectively. Adapted from Ref. [3].

**Figure 5 ijms-24-13565-f005:**
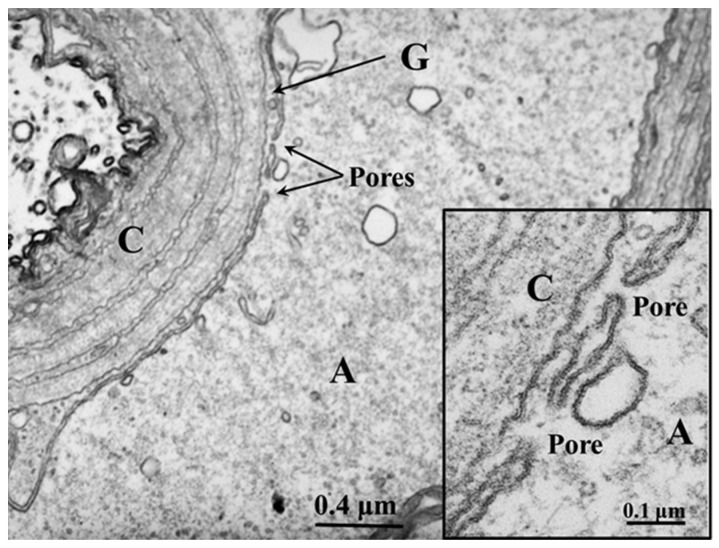
Thin section of a crayfish axon. Axonal and glial cell plasma membranes fuse, forming two pores. The inset is an enlargement of the pore region. A—axon; G—sheath glial cell; C—connective tissue. Adapted from Ref. [7].

**Figure 6 ijms-24-13565-f006:**
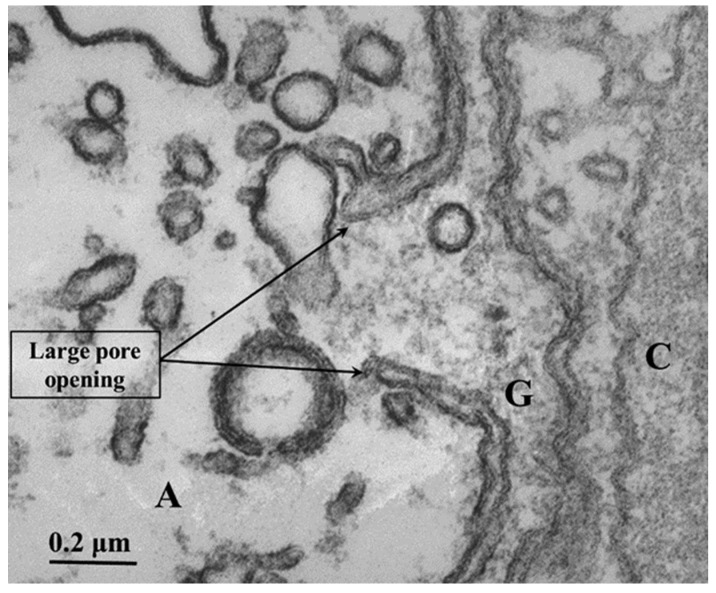
Thin section of crayfish axon displaying a large pore opening. Cytoplasmic organelles are accumulated near the pore region. A—axon; G—sheath glial cell; C—connective tissue. Adapted from Ref. [7].

**Figure 7 ijms-24-13565-f007:**
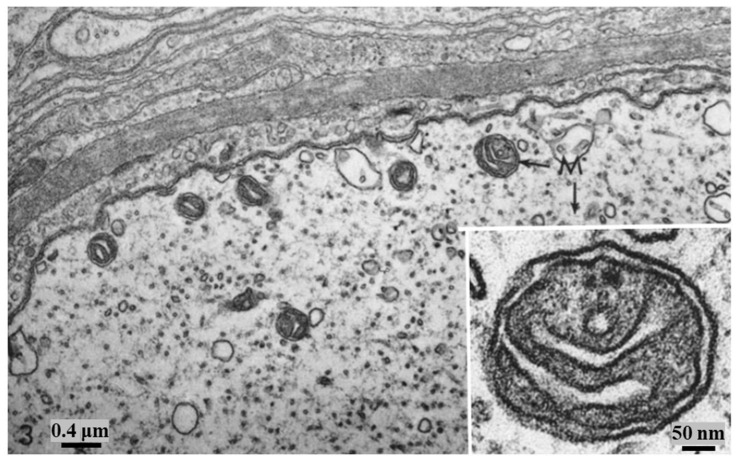
Control axon. Small area of cross-sectioned medium-sized axon. All the axonal membranes display similar electron opacity. The inset shows a mitochondrion (M) at higher magnification; note that both inner and outer mitochondrial membranes display similar thickness and osmiophilia. Adapted from Ref. [2].

**Figure 8 ijms-24-13565-f008:**
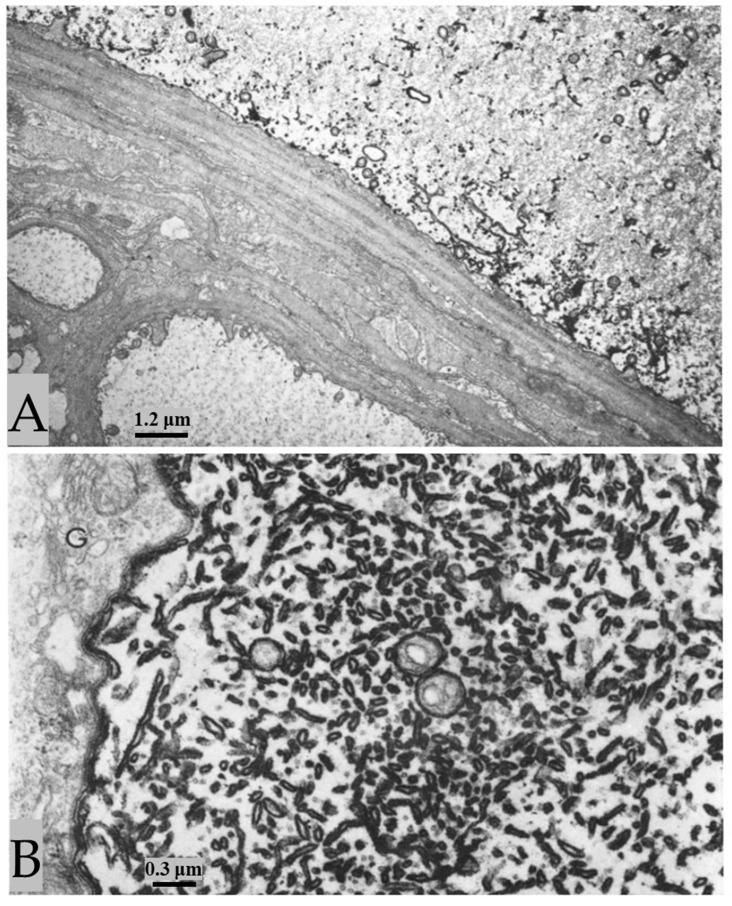
The membranes of lateral giant axons fixed during electrical stimulation display a drastic increase in osmiophilia ((**A**), top axon, and (**B**)), while small neighboring axons show normal osmiophilia ((**A**), left, and bottom axons). Increased osmiophilia is seen in the plasma membrane, the outer membrane of the mitochondria, and the membrane of ER/Golgi cisternae. The lateral giant axon and the two small axons are separated by sheath glial cells and connective tissue (**A**). (**B**), shows an area of a lateral giant axon particularly rich in ER/Golgi cisternae. Note that the mitochondrial outer membranes display very pronounced electron opacity, while the inner mitochondrial membrane and the membranes of the sheath glial cells are as in control axons (**B**). G—Golgi apparatus of the sheath glial cell. Adapted from Ref. [2].

**Figure 9 ijms-24-13565-f009:**
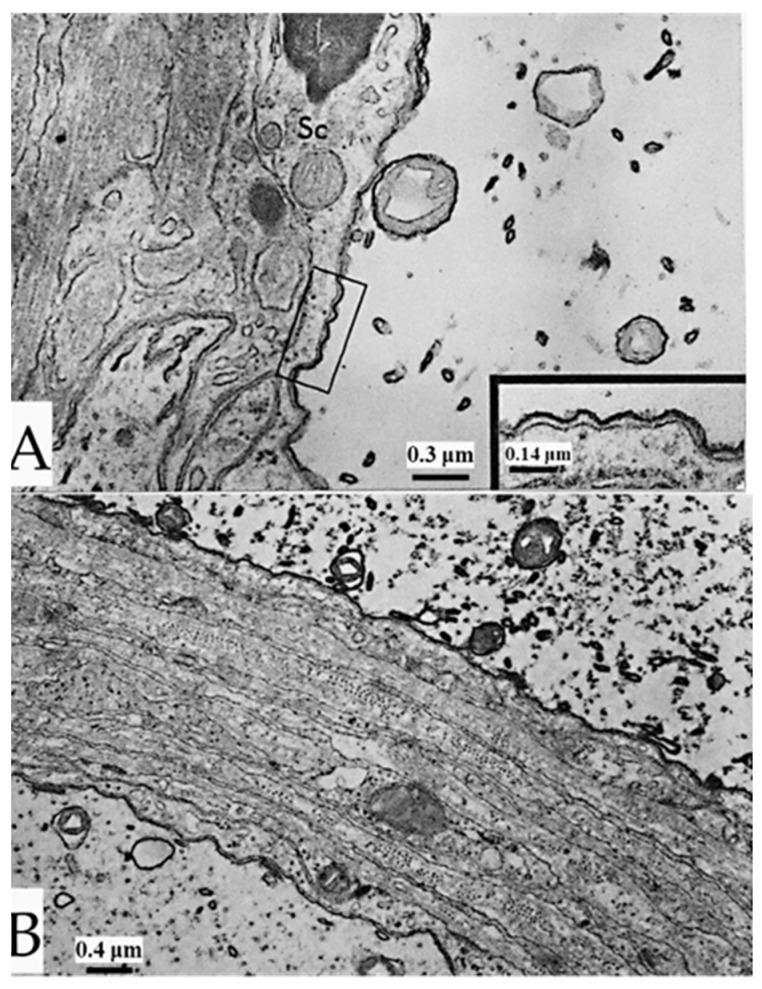
A stimulated median giant axon (**A**) displays increased osmiophilia in the plasma membrane, the outer mitochondrial membrane, and the ER/Golgi membranes, while the inner mitochondrial membrane (**A**) and the membranes of the sheath glial cells ((**A**) and inset) are as in controls. In (**B**), only the top of these two medium-sized axons displays membranes of increased osmiophilia and thickness; this indicates that the reaction occurs only in certain axons (probably the stimulated ones). Adapted from Ref. [2].

**Figure 10 ijms-24-13565-f010:**
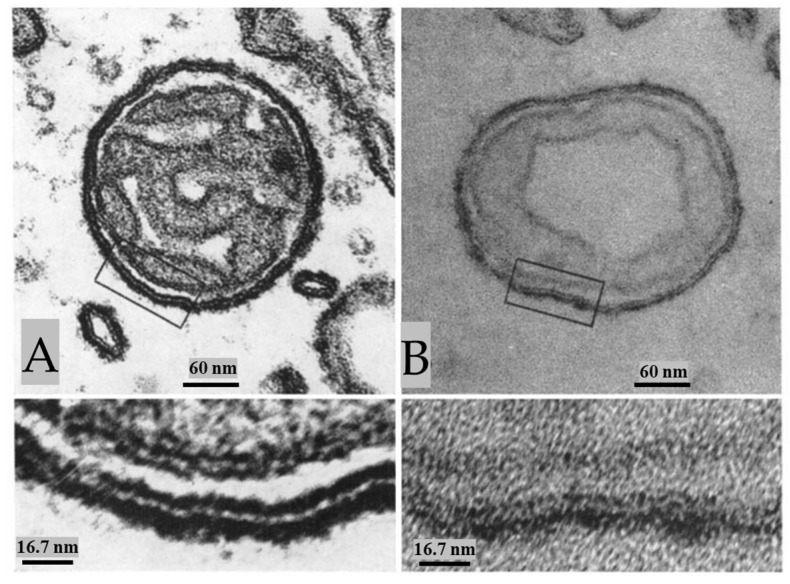
Stimulated axons. In axonal mitochondria stained with uranyl acetate and lead citrate (**A**), the electron-dense leaflets of the outer membrane display increased osmiophilia and thickness, which is more pronounced in the outer leaflet ((**A**), inset). The thickness of the outer membrane is 12–15 nm ((**A**), inset). The increased osmiophilia and thickness are also seen in unstained preparations (**B**). Here also, the axoplasmic leaflet of the membrane is thicker than the inner one ((**B**), inset). Adapted from Ref. [2].

**Figure 11 ijms-24-13565-f011:**
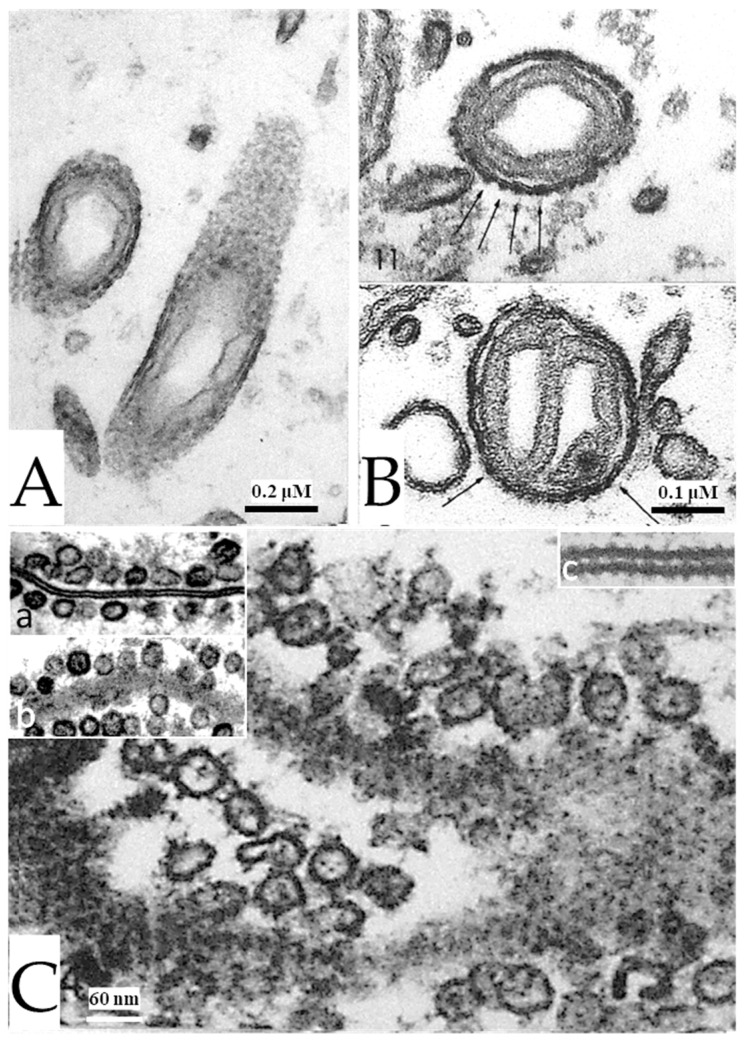
Stimulated axons. In axonal mitochondria obliquely sectioned (**A**), the outer membrane displays granularity, suggesting that the electron-opaque material is not a homogeneous continuous layer. Cross-sectioned outer mitochondrial membranes (**B**) show that the granularity is due to the presence of particles ~15 nm in size spaced every ~20 nm (arrows). In stimulated ganglia, gap junctions of lateral giant axons (**C**) also display increased electron opacity in their typical hexagonal array of cell–cell channels. The channels are spaced at ~20 nm (**c**). Note that the channels are not seen in unstimulated axons fixed with glutaraldehyde/osmium tetroxide (**a**,**b**). Adapted from Ref. [2].

**Figure 12 ijms-24-13565-f012:**
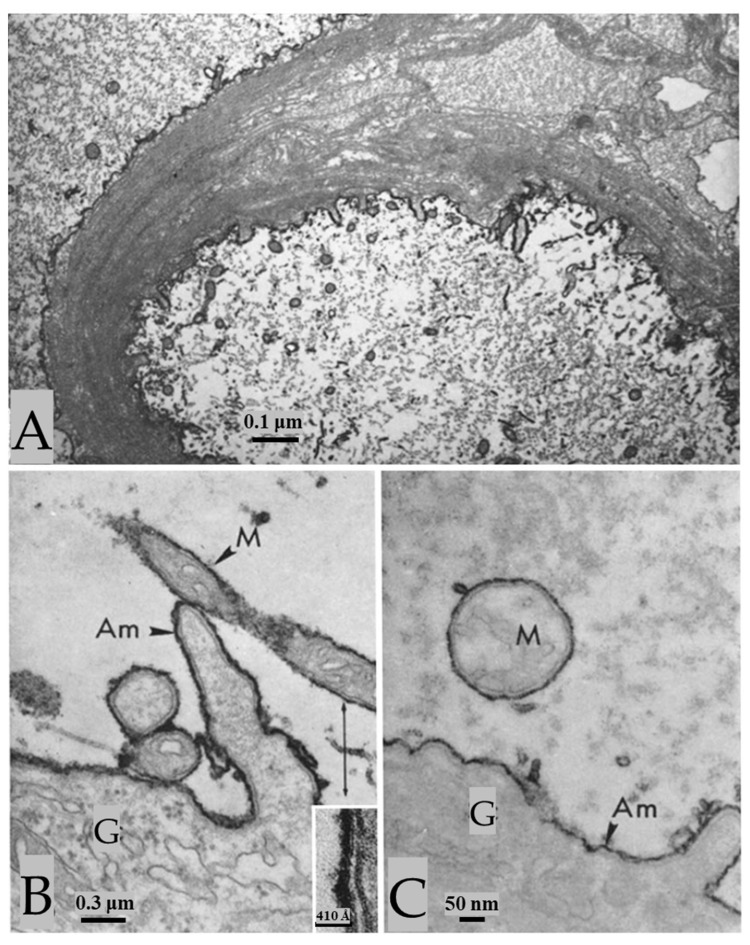
In axons treated with thioglycolate (**A**–**C**) or DTE, all the axons display increased electron opacity in their plasma membranes, outer mitochondrial membranes, and ER/Golgi membranes, as in electrically stimulated (Figure 8, Figure 9, Figure 10 and Figure 11) or asphyxiated axons. In contrast, the inner mitochondrial membrane (**B**,**C**) and the membranes of the sheath glial cells (**B**,**C**) do not show changes. The inset in (**B**) shows inner and outer mitochondrial membranes at higher magnification. Am—Axoplasmic membrane; M—mitochondrion; G—sheath glial cell. Adapted from Ref. [2].

**Figure 13 ijms-24-13565-f013:**
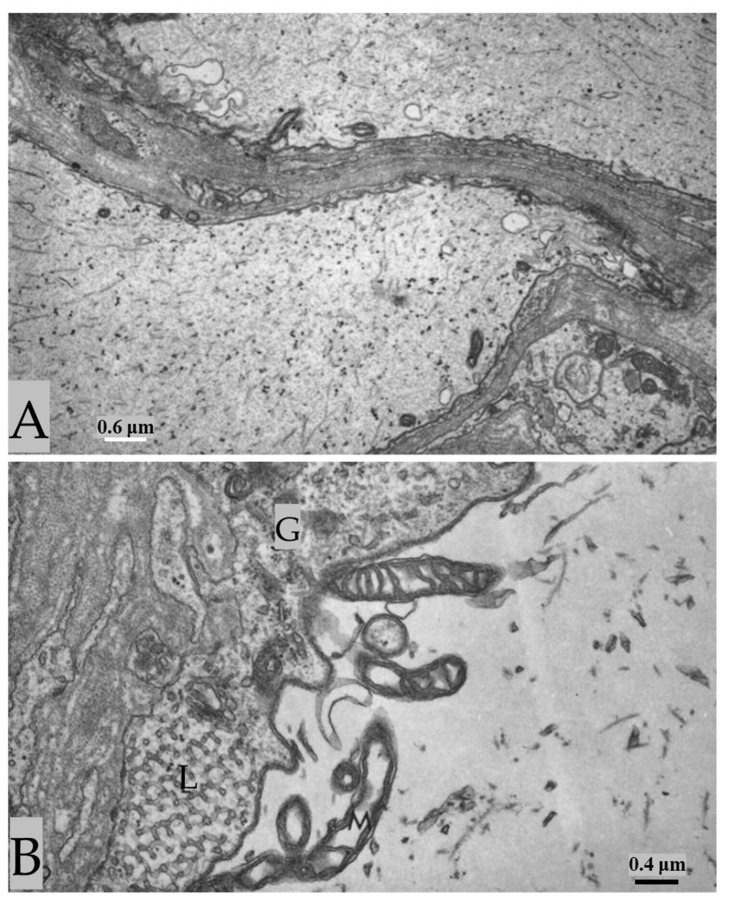
Axons treated with thioglycolate (TG) followed by maleimide. Sulfhydryl groups (-SH), unmasked by TG, are blocked by maleimide (**A**,**B**) of NEM, such that all the membranes display the same electron opacity as control membranes. (**B**), shows a median giant fiber at higher magnification. In the cytoplasm of the sheath glial cell (**B**), one sees the tubular lattice (L), a structure that enables rapid exchange of molecules between the axonal surface and the connective tissue. G—sheath glial cell. Adapted from Ref. [2].

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
