# Peer review of "Nerve Structure-Function: Unusual Structural Details and Unmasking of Sulfhydryl Groups by Electrical Stimulation or Asphyxia in Axon Membranes and Gap Junctions"

_ijms, 2023, doi:10.3390/ijms241713565_

Round 1

Reviewer 1 Report

There are a few unclear structures or descriptions in the text, as marked 'likely':

Line 13 and Line 56 : likely to be kinesin is that without proof ?

Line 14: FS is 'believed'

L19: intercellular pores?  no barrier between glia cells and neurons?

In Figure 5 page 6, could the author identify with a legend the structure on the left circumscribed by the C?

Line 371: what is a 'microsome' in a cell fixed, it is a concept of biochemistry when cells produced extracted membrane pieces in centrifugation density.

line 375: what is re-duction?

Throughout the manuscript, the Angstrom is used, why not nm?

there are references more recent about osmium and fixation... i.e. in Gilloteaux J and Naud J 1979 Histochemistry 63: 227-243 where Ca++ is also invoked, besides -SH groups or peculiar amino acid groups.  In fact, the manuscript is a review but seems to not mention any other component, including Ca++

The review is quite interesting but it should hypothesize about these gap in membrane, in terms of physiology of the axon, what do they meant for future investigations, regarding these and other nervous structures.

Author Response

ANSWERS TO REVIEWER #1

The author is grateful for the helpful comments of the reviewer and has modified the paper as closely as possible to his/her comments.

There are a few unclear structures or descriptions in the text, as marked 'likely': “Likely” is now changed to “probably”

Line 13 and Line 56 : likely to be kinesin is that without proof ? Yes. It is still hypothetical. So, the sentences have been replaced with: “thin filaments, that we believe are made of kinesin” and with:” that we believe to be kinesins [5]. although this needs to be proven”

Line 14: FS is 'believed' The sentence at line 137 0on Fenestrated septa. The word “septa” is plural (latin). So, we need to use “FS are believed”

L19: intercellular pores?  no barrier between glia cells and neurons? The paragraph staring at line 130 explains our view on the nature of the pores: “Pores of this size would be expected to allow wide communication between axoplasm and sheath-glial cell cytoplasm. Indeed, the existence of pores between axons and sheath-glial cells seems reasonable because only passageways of this size would allow the exchange of molecules such as RNA and proteins as heavy as 200,000 Daltons. Probably, the pores form transiently, but are unlikely to be a preparation artifact because the radius of membrane fusion at the pores (Figures 5 and 6) is consistent with the minimum curvature-radius of biological membranes, and the axons’ membranes and other cellular components were very well fixed by vascular perfusion with glutaraldehyde followed by osmium tetroxide”

In Figure 5 page 6, could the author identify with a legend the structure on the left circumscribed by the C? C was already mention in the legend as “C, connective tissue”

Line 371: what is a 'microsome' in a cell fixed, it is a concept of biochemistry when cells produced extracted membrane pieces in centrifugation density. Changed as:” in crustacean nerve microsomes (ER fragments)”

line 375: what is re-duction? Corrected “reduction”

Throughout the manuscript, the Angstrom is used, why not nm? All changed in nanometers “nm”

there are references more recent about osmium and fixation... i.e. in Gilloteaux J and Naud J 1979 Histochemistry 63: 227-243 where Ca++ is also invoked, besides -SH groups or peculiar amino acid groups.  In fact, the manuscript is a review but seems to not mention any other component, including Ca++. Done

The review is quite interesting but it should hypothesize about these gap in membrane, in terms of physiology of the axon, what do they meant for future investigations, regarding these and other nervous structures.

See above: paragraph starting at line 130.

Also, the following lines have been added:” Perhaps, future investigation should at first determine if vertebrate myelinated and/or unmyelinated axons also develop pores under certain conditions, possibly by injecting radioactive tracers or fluorescent dyes in a neuronal cell body and see if the tracer appears in the Schwann cells, as done in crayfish [12]”

Reviewer 2 Report

Please find my comments attached.

Author Response

ANSWERS TO REVIEWER #2

The author is grateful for the helpful comments of the reviewer and has modified the paper as closely as possible to his/her comments.

This is an interesting review, bringing back to light unusual structural details of crayfish axons and changes in osmiophilia of axonal membranes caused by electrical stimulation, asphyxia, or treatment with sulfhydryl reagents. This review is very relevant to stimulate more research on the functional meaning of these early findings. However, the manuscript needs improvement before it can be accepted for publication, since recent papers about mechanical/structural changes occurring in nerves during the action potential discussing shortcomings and possible extensions of the classical Hodgkin-Huxley model are not mentioned in Chapter 3.

Notably, the increase in membrane thickness with electrical stimulation is also found for some organelles (ER and outer mitochondrial membranes) and in gap junctions - which has not been described before - while the inner membrane of mitochondria and the membranes of the sheath-glial cells are not affected. The manuscript presents the hypothesis that electrical stimulation causes unmasking of -SH groups in excitable membranes which might lead to conformational changes in proteins. Zuazaga de Ortiz and del Castillo suggested that the appearance of excitability in crustacean muscle depends on the conversion of certain CH2-SH side chains to thioethers with carbonyl groups, these new carbonyl groups may interact with neighboring amino groups and form bonds between different regions of a protein or between different protein units, which might affect excitability by inducing conformational changes or by preventing the occurrence of them (PMID: 700016). Can the author comment on this?

The paper by Zuazaga de Ortiz is now cited. This scenario should definitely be kept in mind, but it might not apply to our data with TG or DTE. See lines 396-403.

Further remarks:

l.40 typo mediun (medium) Done

l.43 typo togerther (together) Done

l.54 In Fig. 2 B, the text box 0.1 m overlaps with C, in Fig. 2 C the text box 0.1 m is cut off on the right side of the image Done

l.65 unit symbol 130-140 Ǻ (Å) Done. Note that based of Rev. #1, all “Å” have been replaced with “nm”

l.81-97 Figure 4 is missing or labeling is wrong in the text 2. I had noticed that, even though the manuscript I submitted contained Fig. 4. I had asked the editor to send to reviewers the proper copy that contained Fig. 4. I was told that it was done. Sorry if you did not receive it.

l.135 Engelbrecht et al. (PMID: 32500424) describe assumptions in modelling including a mechanical (pressure) wave in the axoplasm generated during the action potential (AP). This and other relevant missing data have been added. See lines 187-191.

l.146 It would be helpful for the interested reader to add references here. An overview about recent theoretical and experimental work is provided in PMID: 33037976 and PMID: 29981394. Fillafer et al. have reported cell surface deformation during AP in plant cells (PMID: 29401438). A paragraph has been added to address these comments. See Lines 150-161

l.161 The question concerning reversibility of heat production in nerves during AP propagation is of central importance and has been addressed in PMID: 32805276 and https://doi.org/10.1515/jnet-2019-0012. There is currently no widely accepted consensus about the mechanism of heat generation (PMID: 32500424). Two sentences have been added. See Lines 187-190

l.180 A rapid volume expansion of fibers Done

l.231 typo thew (the) Done

l.250 increased electron density Done

l.345 the membranes of the ER/Golgi cisternae Done

l.347 whose -SH groups (that) are unmasked by electrical stimulation Done. Line 363

l.347-50 Indeed, Baumgold et al. suggested that sulfhydryl groups, located on the axoplasmic side of the axolemma, are involved in the maintenance of excitability in the sense that modification of these groups produces conduction block (PMID: 621524). Done. See Lines 369-372

l.357 An ion channel redox model was brought forward in 1991 by Marinov (PMID: 1709974), suggesting thiols in oxidized and reduced form as redox centers.

l.359 reduction Done. See Lines 379-382

l.367 on the functional meaning of these early findings. Ref. added “functional meaning our early findings

Round 2

Reviewer 2 Report

The manuscript has been greatly improved, but there are some remaining issues that need to be addressed, as listed below:

l. 208  rapid volume expansion of fibers 

l. 353/4  Of note, Ca2+ is necessary for nerve impulse activity and plays an important role in the mechanism determining the specific conductance changes in the nerve membrane, as described by Frankenhaeuser, 1957 (PMID: 13449875). Furthermore, a redox-dependent thiol switch might affect a nearby Ca2+ binding site (PMID: 35605898), and it has been proposed that structural changes induced by disulfide bond formation may interfere with Ca2+ binding leading to functional consequences such as enzyme inactivation (PMID: 25404341). 

l. 381/2  This model suggests that thiols in oxidized and reduced form may act as redox centers

l. 385 there are proteins rich in sulfur whose -SH groups are unmasked by electrical stimulation

l. 392/3  Regarding the role of SH groups in ATPases see discussion in Lin and Ayala [Lin S., Ayala G.F. Effect of sulfhydryl group reagents on the crayfish stretch receptor neuron. Comp. Biochem. Physiol. Part C Comp. Pharmacol. 1983;75:231–237. doi: 10.1016/0742-8413(83)90186-X]. These authors suggest that protein conformational changes occurring when free SH groups are bound by SH reagents would very likely involve either the release of membrane-bound Ca2+ (Tolberg & Maccy, 1972) or the inhibition of membrane binding of Ca2+ (Palmer & Posey, 1970), resulting in altered membrane function (cf. Treherne, 1966).

l. 403  by treatments with TG or DTE

l. 405  reduction of disulfide bonds

Author Response

The author is grateful for the latest comments of the reviewer and has modified the paper as closely as possible to his/her comments.

  1. 208  rapid volume expansion offibers. Done
  2. 353/4  Of note, Ca2+ is necessary for nerve impulse activity and plays an important role in the mechanism determining the specific conductance changes in the nerve membrane, as described by Frankenhaeuser, 1957 (PMID: 13449875). Furthermore, a redox-dependent thiol switch might affect a nearby Ca2+ binding site (PMID: 35605898), and it has been proposed that structural changes induced by disulfide bond formation may interfere with Ca2+ binding leading to functional consequences such as enzyme inactivation (PMID: 25404341). Done
  3. 381/2  This model suggests that thiols in oxidized and reduced form mayact as redox centers Done
  4. 385 there are proteins rich in sulfur whose -SH groups are unmasked by electrical stimulation. Done
  5. 392/3  Regarding the role of SH groups in ATPases see discussion in Lin and Ayala [Lin S., Ayala G.F. Effect of sulfhydryl group reagents on the crayfish stretch receptor neuron. Comp. Biochem. Physiol. Part C Comp. Pharmacol. 1983;75:231–237. doi: 10.1016/0742-8413(83)90186-X]. These authors suggest that protein conformational changes occurring when free SH groups are bound by SH reagents would very likely involve either the release of membrane-bound Ca2+ (Tolberg & Maccy, 1972) or the inhibition of membrane binding of Ca2+ (Palmer & Posey, 1970), resulting in altered membrane function (cf. Treherne, 1966). Done
  6. 403  by treatmentswith TG or DTE. Done
  7. 405  reduction of disulfide bonds. Done